# Clinical Spectrum and Outcomes of Cryptogenic *Klebsiella pneumoniae* Liver Abscess in the Americas: A Scoping Review

**DOI:** 10.3390/pathogens12050661

**Published:** 2023-04-29

**Authors:** Jorge Cardenas-Alvarez, Galit Balayla, Abel Triana, Rodrigo Diaz Lankenau, Carlos Franco-Paredes, Andrés F. Henao-Martínez, Gabriel Motoa

**Affiliations:** 1Department of Medicine, Jackson Memorial Hospital, University of Miami, Miami, FL 33136, USA; tabel@uninorte.edu.co (A.T.); rodrigo.diazlankena@jhsmiami.org (R.D.L.); gxm813@med.miami.edu (G.M.); 2Department of Medicine, Icahn School of Medicine at Mount Sinai Morningside-West, New York, NY 10019, USA; galit.balaylarosemberg@mountsinai.org; 3Hospital Infantil de México Federico Gomez, Mexico City 06720, Mexico; carlos.franco.paredes@gmail.com; 4Department of Microbiology, Immunology and Pathology, Colorado State University, Fort Collins, CO 80523, USA; 5Division of Infectious Diseases, University of Colorado, Anschutz Medical Campus, Aurora, CO 80045, USA; andres.henaomartinez@cuanschutz.edu

**Keywords:** liver abscess, America, *Klebsiella pneumoniae*, cryptogenic

## Abstract

(1) Background: Cryptogenic *Klebsiella pneumoniae* liver abscesses are an invasive infection with or without extra hepatic involvement in the absence of hepatobiliary disease or abdominal malignancy. Most of the evidence has emanated from reports from Asia, and previous studies in the Americas have limited clinical characterization. (2) Methods: To understand this syndrome’s characteristics on our continent, we conducted a scoping review to identify adult cases of idiopathic, community-acquired monomicrobial *K. pneumoniae* liver abscess in the Americas. (3) Results: We identified 144 cases spanning 1978–2022. Most cases were reported in males that had traveled or migrated from Southeast or East Asia with diabetes mellitus. Extrahepatic involvement and bacteremia were common, including seeding to the lungs, ocular structures, and central nervous system. Although limited by sample size, the most commonly reported genes were *magA* or *rmpA*. Concomitant percutaneous drainage and third generation cephalosporins (alone or in combination with other antibiotics) were frequently used, yet pooled fatality occurred in 9% of the reported cases. (4) Conclusions: The features of cryptogenic *K. pneumoniae* liver abscess in the Americas mirror those described in Asia, confirming its global dissemination. This condition is increasingly being reported in our continent and carries significant clinical impact due to its systemic invasiveness.

## 1. Introduction

*Klebsiella pneumoniae* is a Gram-negative pathogen with two distinctive strains (traditional and hypervirulent), both capable of causing a myriad of infectious syndromes, including liver abscess. [1,2]. Hypervirulent *K. pneumoniae* strains are evolving virulent pathogens with unique features, including the propensity to cause community-acquired abscesses inside and outside of the liver in relatively healthy individuals of any age group. Unlike the traditional *K. pneumoniae* strain, hypervirulent *K. pneumoniae* tends to disseminate to other organs such as the eye, the central nervous system, or the lungs, and is associated with increased mortality and poor clinical outcomes [3,4,5,6]. This infection, now referred to as cryptogenic *K. pneumoniae* liver abscess (cKLA), has been predominantly linked to the hypervirulent form. It was initially described in Taiwan in 1986 [7], but increasing evidence has demonstrated its global spread [4]. The terms cryptogenic, primary, or idiopathic have been used to describe these liver abscesses when they occur in the absence of previous abdominal surgery or concomitant intra-abdominal pathology (i.e., colorectal malignancy or previous hepatobiliary disease). Historically, these pyogenic liver abscesses have been reported as polymicrobial, with *E. coli* causing most cases. However, in the last three decades, the frequency of cKLA has increased around the globe, particularly in Asia, where it is now the most common etiology [6]. Additionally, they are encountered more commonly in diabetic—yet otherwise immunocompetent—males originally from the Asian Pacific Rim [8,9,10,11].

Although strong emphasis is made on liver abscess, hypervirulent *K. pneumoniae* may cause a variety of entities with and without multiple sites of infection outside of the liver, as described in a case of renal abscess with metastasis to the lungs, eye, and brain reported in Dallas, USA [12].

The genomic analysis of the most common strains is associated with various genes, including the mucoviscosity-associated gene A (*magA*, now called *wzy_KpK1_*), the regulator of mucoid phenotype A gene (*rmpA/A2)*, the salmochelin siderophore biosynthesis gene (*iroB)*, aerobactin siderophore biosynthesis gene (*iucA)*, and others [12,13,14,15]. These mediate the virulence and mucoid phenotype seen in the K serotypes (mainly K1, K2, but also K5, K20, K30, K25, K27, K32, K57, etc.) or others by the production of a hypercapsule, a variety of extracellular saccharides (including colonic acid), and excessive siderophores [13,16,17,18,19,20].

Although in the past decades, cKLA has risen as a unique infection worldwide, comprehensive clinical and epidemiological information in the Americas is limited. Unfortunately, the available literature consists of case reports and case series, and it neglects cases outside of the United States of America (USA). The two largest reports of cases of *K. pneumoniae* liver abscess in the USA were conducted by Siu et al. [21] and Fazili et al. [22]. They described 38 and 93 cases, respectively. However, they included individuals with hepatobiliary disease, which may inherently predispose individuals to a pyogenic liver abscess; neither included countries other than the USA.

In this study, we aim to describe a scoping review of cases reported in the American continent by delineating the clinical and epidemiological features of this syndrome, and we aim to further characterize the clinical outcomes to increase awareness and guide management.

## 2. Materials and Methods

This scoping review was conducted following the Joanna Briggs Institute Methodology for Scoping Reviews and was reported according to the Preferred Reporting Items for Systematic Reviews and Meta-analyses Extension for Scoping Reviews (PRISMA-ScR) [23,24].

### 2.1. Study Eligibility and Search Strategy

The literature search strategy involved searching the articles in databases that included PubMed/Medline, Embase, Scopus, Web of Science, LILACS, and CINAHL. The cross-searching was conducted based on three categories: (1) Liver; (2) *Klebsiella pneumoniae* related terms; (3) Abscess. The detailed search strategy, according to each database, is presented in the Appendix A. Two investigators independently conducted the literature search (AT, JC). We searched the databases from their inception to July 2022, and only publications written in English, Spanish, or Portuguese were included. We used text words and medical subject heading (MeSH) terms using the Boolean search strategy. Our literature search was supplemented by snowball search in databases such as Google Scholar, Latin American and Caribbean Health Sciences Literature (LILACS), and Open Grey.

Only case reports, case series, abstracts, and letters to the editor were included in our search, as no higher-level evidence studies were found for this specific research question. To be eligible, the studies should have described cases of monomicrobial liver abscess due to *K. pneumoniae* confirmed by blood or surgical specimen, reported in the American continent (North, Central, or South America), in adults (≥16 years old). We excluded studies that evaluated patients with active hepatobiliary disease, concurrent intra-abdominal malignancy, or abdominopelvic surgery in the past 3 months before presentation, and cases that failed to meet the inclusion criteria. In vitro studies, studies on experimental animal models, and case series unable to individualize data and only presented as ranges or averages were excluded. The following results were imported to the desktop version of Covidence systematic review software to merge duplications and prepare for the screening. In addition, the gray literature was reviewed to ensure no information was left out.

### 2.2. Study Selection and Data Extraction

Two authors (AT, JC) screened results by titles and abstracts identified with the search strategy presented. After the screening, the full-text preselected articles were examined by two authors (AT, JC) to determine if they met the inclusion and exclusion criteria. Any discrepancy among the authors was settled through discussion until a consensus was reached. The list of articles included in our study is presented in Appendix B.

A data abstraction form was developed, and all included studies were abstracted by two reviewers (RD, JC) working independently. Discrepancies were resolved through discussion. Data items included study characteristics (author’s name, year, and country of publication), patient characteristics (age, gender, country of origin, clinical presentation, history of travel, comorbidities, phenotypic and genotypic profile of isolates, treatment, and duration of antibiotics), and outcomes (presence of extrahepatic disease, cure rates, relapse, and mortality).

### 2.3. Data Synthesis and Statistical Analysis

Continuous variables were summarized with the sample median and range. Categorical variables were summarized with the number and percentage of patients and compared using Chi-square tests or Fisher’s exact test. The *p* values < 0.05 were considered statistically significant. All statistical tests were 2-sided. Statistical analyses were performed using SAS 9.2 (SAS Institute, Cary, NC, USA).

### 2.4. Study Definitions

Country of diagnosis was defined as the country of origin of the first author. Region of origin was defined as the geographic region where the patient was originally from, and it should have been explicitly mentioned in the text. Vision complaints were defined as any of the following: eye discomfort, eye redness, change in visual acuity, or eye discharge. Any change in the baseline from visual acuity after infection (including enucleation) was defined as sequelae. Sepsis was defined according to the syndrome of inflammatory systemic response criteria with confirmed infection [25]. A string test was considered positive when *K. pneumoniae* colonies were stretched in culture media using an inoculation loop, and the resulting viscous string was ≥5 mm in length [26].

## 3. Results

A total of 6262 citations resulted from our search strategy. After removing duplicates, 2806 abstracts and titles were screened. Sixteen additional studies were found through other sources. The full-text review was conducted on 167 articles, of which 43 were excluded (reasons for exclusion are displayed in Figure 1). A total of 124 studies were included in the analysis.

### 3.1. Study Characteristics

Included studies were published between 1978 and July 2022. The number of cases per decade occurred as follows: 1978–1979 (*n* = 1), 1980–1989 (*n* = 1), 1990–1999 (*n* = 8), 2000–2009 (*n* = 27), 2010–2019 (*n* = 81), 2020–2022 (*n* = 26), showing a definitive trend up by decade. The distribution of cases reported per country is summarized in Figure 2. All studies were anecdotal case reports or case series, with some presented in the form of a meeting abstract.

### 3.2. Population Characteristics

A total of 144 cases of cKLA were identified in our cohort. A pooled analysis showed a median age of 52 (range: 18–86) years old, predominantly male, and who originally were from countries located in Southeast and East Asia. The most commonly reported symptoms were fever (84%), abdominal pain (54%), and constitutional symptoms (49%). A summary of the population characteristics, ethnicity, and genotypic and phenotypic profiles is displayed in Table 1.

History of travel within three months of diagnosis was described in nine patients (seven were diagnosed in the USA and two in Canada), and the countries of travel were the Philippines (*n* = 4), Mexico (*n* = 2), Algeria (*n* = 2), and Vietnam (*n* = 1).

Antibiotics were reported as part of the therapy in 104 cases (72.22%). It is unclear if the rest did not receive antimicrobials or if the article failed to mention it. The duration of therapy was only reported in 61 (42.36%) cases. The median duration of therapy was six (range: 2–6) weeks. The antibiotic regimen was highly heterogeneous, yet the most frequent agent used, either alone or in combination, was ceftriaxone in 47 cases (32.64%). One hundred and sixteen cases (80.56%) underwent percutaneous drainage, thirteen (9.03%) underwent surgical drainage, and sixteen (11.11%) had no drainage at all.

### 3.3. Outcomes

Bacteremia was reported in 59.72% of the cases. Extrahepatic disease was reported in 82 individuals (56.94%) and was described as follows: brain or meningeal involvement (10.42%), eye (22.92%) (most frequently endophthalmitis in 86.36% of the cases, followed by chorioretinitis, panophthalmitis, uveitis, or subretinal abscess), lung or pleura (29.17%) (includes pneumonia, bronchopneumonia, lung abscess, pleural effusion, or empyema), other abdominal/pelvic organs (11.11%) (including adrenal, renal, prostatic, or psoas muscle), skin and soft tissue (4.17%) (including necrotizing fasciitis), and osteoarticular (including septic arthritis, and osteomyelitis) (2.08%). At the time of diagnosis, three or more organs and systems were simultaneously involved in 23 individuals (15.97%). Of note, all patients with eye disease had eye sequelae described, and 18/33 were reported to have sepsis in the case description.

Total or partial resolution of the abscess were recorded in 87/97 patients (89.69%). Relapsed liver abscess after treatment was recorded in 1/12 studies (8.33%). Both cure rates and relapse were not mentioned in 47 and 132 studies, respectively. The mortality in our study was 9.03%. Among the thirteen reported deaths, all patients but one had some form of extrahepatic disease, eight had diabetes mellitus (61.54%), eight had bacteremia (61.54%), and two had ocular disease (15.38%).

## 4. Discussion

This study describes the clinical, epidemiological, and microbiological features of cKLA in the American continent. Most patients reported were middle-aged males, predominantly from Southeast and East Asia, with diabetes as the most common comorbidity (40.28%). These findings closely mirror the epidemiology previously described in Asia, where the disease is now considered endemic [27,28,29]. Both *magA* and *rmpA* genes and K1 serotypes played a crucial role in the pathogenesis of our population, which coincides with the findings described by Fung et al. [30]. All these clinical and genetic similarities reinforce that cKLA is actively spreading to other regions of the world.

The spread occurs most likely through the fecal–oral route, since *Klebsiella* spp. capsular antigens are known to facilitate intestinal colonization, and the clonal dissemination of K1 serotypes has already been described in Korea [31,32]. Nonetheless, the definitive route of entry is still not well described. It remains uncertain if the pathogen disseminates through the systemic circulation and seeds in the liver, or if, conversely, the liver is the primary source of infection (due to translocation from the gastrointestinal tract) from where it disseminates to the blood and other organs, or if it is a combination of both. Regardless, no evidence of gastrointestinal, respiratory or urinary colonization of *Klebsiella* spp. were reported in our cohort.

Extrahepatic disease was common in our study (56.94%), which has significant implications for the management and monitoring of seeding. Searching for occult abscesses should be considered, as it may impact antimicrobial choice and duration and source control strategies.

One of the most alarming findings of this study is how abysmal the prognosis can be after eye involvement (including endophthalmitis, chorioretinitis, uveitis, or subretinal abscess). Virtually all individuals had documented eye sequelae in our cohort—this has been similarly reported by other studies [33,34]. These findings suggest that prompt consultation with an ophthalmologist should be considered in this population.

It is remarkable how Asians are disproportionately affected by hypervirulent *K. pneumoniae* strains, even outside endemic areas. Although we do not know the exact travel conditions in our cohort, over half of the patients had either traveled recently or were originally from the Asian Pacific Rim. While it is difficult to draw definitive conclusions because of the nature of this study, the small sample size, and the data quality, this pattern suggests a higher risk among the Asian population in the Americas. A mixture of environmental and host-related factors, including increased intestinal carriage of hypervirulent *K. pneumoniae* strains and inherent genetic predisposition, are believed to place this particular group at higher risk for cKLA [28,35,36,37,38,39]. Despite this predisposition, our study shows that non-Asian individuals (Hispanic/Latino, African American, White/Caucasians, Pacific Islanders, etc.) are susceptible even without a history of travel to endemic areas.

Diabetes and hypertension were the most frequent comorbidities in our study (40.28% and 13.89%, respectively). Mukherjee et al. found similar frequencies of diabetes (60% vs. 56.3%) and hypertension (46.7 vs. 50.7%) in patients with metastatic and non-metastatic cKLA in a study in Singapore between 2013 and 2017. Although diabetes has been associated with metastatic complications in other studies, it was not found to be a risk factor for metastatic infection in that cohort [40]. Other reports have described Southeast/East Asian origin, male gender, symptomatic disease (dyspnea or sepsis), and abscess size as predictors for metastatic complications [30,40,41,42].

The genotypic profile in our study re-demonstrates the characteristics described in other populations [43]. Understanding these features is crucial, as their application in clinical practice and epidemiology is currently evolving. Genotypic biomarkers (e.g., *iroB*, *iucA*, *rmpA*, and *rmpA2*) have been described by Russo et al. to predict the presence of hypervirulent strains accurately. Yet, its use has yet to be universally validated [44]. Routine use of the string test in the microbiology laboratory should be cautiously interpreted, as it has variable sensitivity and poor positive predictive value for hypervirulent *K. pneumoniae* [45,46].

Antibiotic therapy is widely variable in cKLA because there is no standard of care. In our cohort, the most common antibiotics used as monotherapy included third-generation cephalosporins (16.67%), piperacillin-tazobactam (9.72%), and fluoroquinolones (7.63%). Based on retrospective analyses, some authors suggest either using a third- or first-generation cephalosporins +/− aminoglycosides (the latter for the first two weeks of therapy), although antimicrobial susceptibility of hypervirulent *K. pneumoniae* tends to be favorable, and other regimens are considered to be equally acceptable [24,47,48,49]. Recently, the emergence of Carbapenem-resistant hypervirulent strains of ST11 lineage have been a public health concern [18]. No cases of carbapenem-resistant strains were recorded in our study. This study’s mean duration of antibiotic therapy was six weeks, but this is solely based on case reports/series.

Percutaneous drainage (aspiration or pigtail) should be performed, if possible. In our study cohort, mortality in the percutaneous group was 4.5% vs. 30% in the non-percutaneous drainage group, although the difference was not statistically significant (*p* 0.3796). Previous research has favored a percutaneous approach [48], which sometimes might be limited due to the viscosity of the abscess, the location, or the collection’s characteristics; thus, a surgical approach might be needed [50].

This study has several limitations, including the quality of the evidence and the small sample size. Most cases were reported in North America, which may represent publication practices in this region in comparison to other locations of this continent. When describing symptoms, fever was the most common symptom, which tends to be more frequently reported than others and may underestimate the prevalence of other symptoms. There is still insufficient evidence to elucidate the pathogenesis of cKLA. Additional studies are necessary to generate and test hypotheses within this topic. Lastly, no laboratory data were collected, which leaves a gap in knowledge regarding this variable.

## 5. Conclusions

In conclusion, the clinical, epidemiological, and microbiological features of cKLA closely mirror those described in Asia, confirming this pathogen’s global dissemination. Rapid identification of this condition is crucial, as the consequences can be severe, including increased mortality and blindness. We used the data in our study to frame a proposed diagnostic and therapeutic approach to cKLA, displayed in Figure 3. These results should raise awareness of cKLA in the Americas; further studies are necessary to elucidate unanswered questions regarding the dynamics of this syndrome in our continent, the standard of care in antibiotic management, and the appropriate approach to those with extrahepatic disease.

## Figures and Tables

**Figure 1 pathogens-12-00661-f001:**
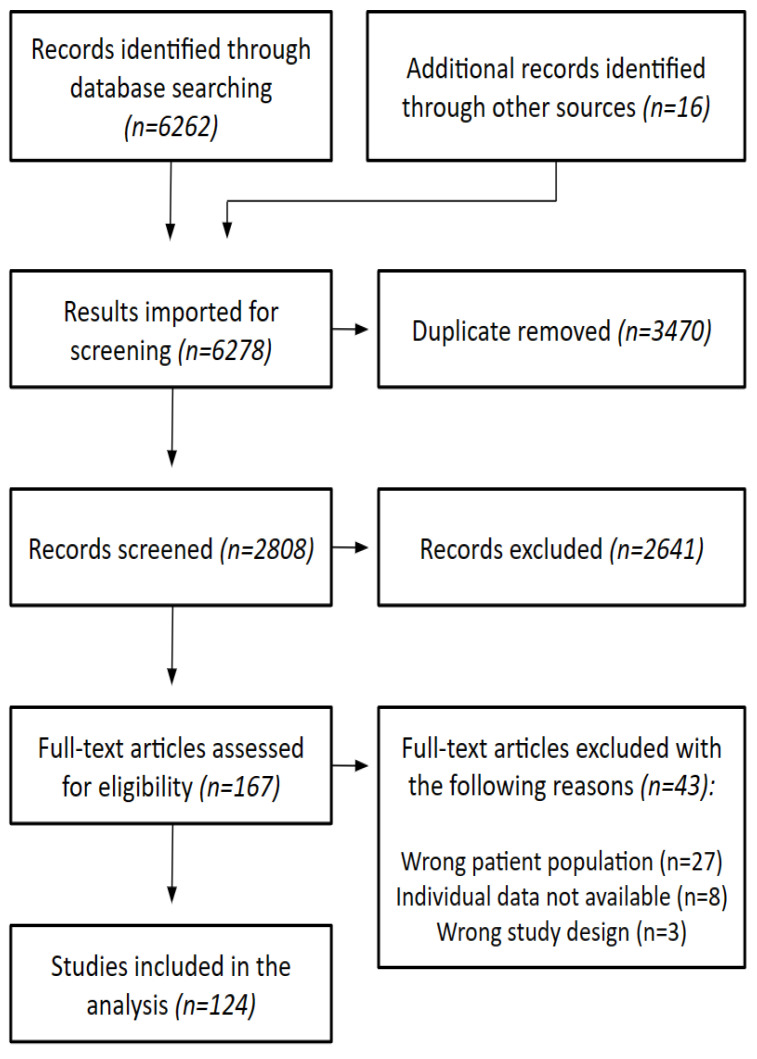
Flowchart of the search, screening, and selection of the evidence.

**Figure 2 pathogens-12-00661-f002:**
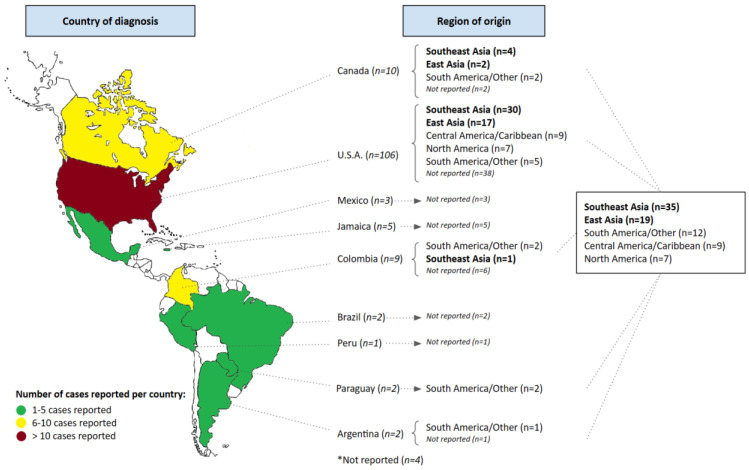
Number of cases reported in the Americas (per country) and their reported origin.

**Figure 3 pathogens-12-00661-f003:**
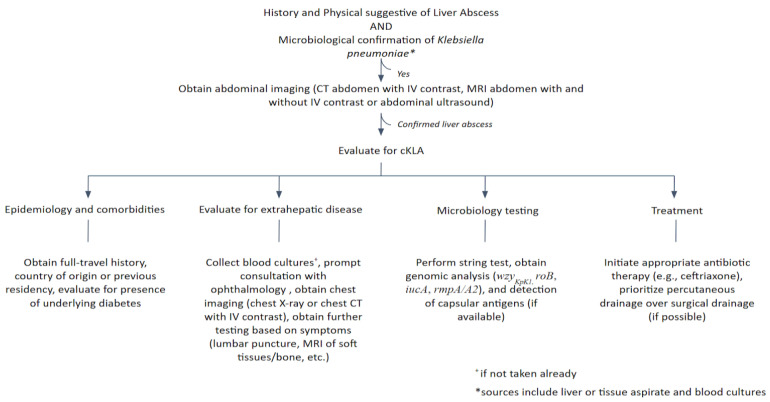
Approach to individuals with confirmed and/or suspected cKLA.

**Table 1 pathogens-12-00661-t001:** Population characteristics.

	Total [n (%)]
**Age median** (range)	52 (18–86) years
**Gender** (*n* = 144)	
Male, n (%)	112 (77.78%)
**Comorbidities** (*n* = 144)	
Diabetes mellitus, n (%)	58 (40.28%)
Hypertension, n (%)	20 (13.89%)
Hyperlipidemia/Coronary artery disease, n (%)	10 (6.94%)
Alcohol use, n (%)	4 (2.77%)
Concomitant malignancy, n (%)	3 (2.08%)
**Reported Symptoms** (*n* = 144)	
Fever, n (%)	121 (84.03%)
Abdominal pain, n (%)	78 (54.17%)
Constitutional symptoms, n (%)	71 (49.31%)
Nausea/vomiting, n (%)	48 (33.33%)
Vision complaints, n (%)	28 (19.44%)
Diarrhea, n (%)	19 (13.19%)
Headache, n (%)	17 (11.81%)
Dyspnea, n (%)	14 (9.72%)
Cough, n (%)	11 (7.64%)
Urinary symptoms, n (%)	11 (7.64%)
Jaundice, n (%)	10 (6.25%)
Chest pain, n (%)	4 (2.77%)
**Ethnicities** (*n* = 144)	
Asian, n (%)	56 (38.89%)
Hispanic/Latino, n (%)	16 (11.11%)
White/Caucasian, n (%)	8 (5.56%)
African American, n (%)	6 (4.17%)
Afro-Caribbean, n (%)	4 (2.78%)
Pacific Islanders, n (%)	3 (2.08%)
Asian American, n (%)	2 (1.39%)
Not reported, n (%)	49 (34.03%)
**Bacteremia** (*n* = 144)	86 (59.72%)
**Extrahepatic disease** (*n* = 144)	
Lungs or pleura, n (%) ^*^	42 (29.17%)
Ocular structures, n (%) ^+^	33 (22.92%)
Other intra-abdominal abscesses, n (%) ^#^	16 (11.11%)
Brain and meninges, n (%) ^ˠ^	15 (10.42%)
**Genotypic profile** (*n* = 19)	
*magA*/*wzy_KpK1_* gene, n (%)	11 (57.89%)
*rmpA* gene, n (%)	13 (69.42%)
*iro*, *iuc*, *ybtA*, *clbA*, and/or *peg-344* genes, n (%)	4 (21.05%)
**Phenotypic profile** (*n* = 32)	
Hypermucoviscosity, n (%)	28 (87.50%)
K1 serotype, n (%)	9 (28.13%)
K2 serotype, n (%)	4 (12.50%)
Non K1/K2 serotype, n (%)	2 (6.25%)
Siderophores, n (%)	0 (0.00%)

* Including pneumonia, bronchopneumonia, lung abscess, pleural effusion, or empyema + including endophthalmitis, chorioretinitis, panophthalmitis, uveitis, or subretinal abscess # including adrenal, renal, prostatic, or psoas muscle ˠ including meningitis or brain abscess.

## Data Availability

The data presented in this study are available on request from the corresponding author.

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
