# Peer review of "Clinical Spectrum and Outcomes of Cryptogenic Klebsiella pneumoniae Liver Abscess in the Americas: A Scoping Review"

_pathogens, 2023, doi:10.3390/pathogens12050661_

Round 1
Reviewer 1 Report
This study is a systematic review of cryptogenic Klebsiella pneumoniae liver abscess (cKLA). This study is well designed, and the results obtained from this study are reliable.
Liver abscess is a serious and important disease, though it is relatively rare. The results of this study can contribute to the control of cKLA, and thus this manuscript should be published in pathogens.
However, the authors should add the below to the discussion of this manuscript.
1. The authors use the term “hypervirulent Kp strains” several times in the discussion. However, the definition of hypervirulent Kp strains is unclear. They should state the definition of hypervirulent Kp strains more clearly.
2. The authors have not fully discussed the relationship between cKLA and background diseases. The authors should discuss it in more detail based on the data in Table 1.
Reviewer 2 Report
Alvarez j et al investigated the problem of hypervirulent K pneumoniae strains causing liver abscesses in North and South America. The topic is interesting.
Comments:
In the title – Suggest authors to write “North and South America” instead “Americas”
Abstract – which “antimicrobials” , please specify the antibiotics (e.g ceftriaxone as it is shown on page 5).
Introduction and also for the entire manuscript – please don’t use word “entity”
Introduction - Please provide explanation what is cryptogenic
Page 2,second paragraph, please include information for the structure that encode these genes – wzy (for capsule). Rmp (regulator of mucoid phenotype), siderophores ….
Page 2 second paragraph – Why K32, serotypes that are associated with hypervirulence are K1, K2 (with K1 and K2 the most important), but also K5, K10, K20, K25, K27, and K57. https://doi.org/10.1002/jcla.24743
In introduction - Please provide some background for pyogenic liver abscesses problem – what is this disease, what was the main causing agent in the past and what is the difference nowadays. https://doi.org/10.3201/eid2402.170957
Please provide more information on the hvKp strains – what disease they are associated with, which serotypes, the importance of Kp producing carbapenemases from the ST11 lineage.
There is a problem with the inclusion criteria – the aim of the paper is to perform meta-analysis of data from Americas, but the abstract says that “We identified 144 cases during 1978-2022. Most cases were reported in males from Southeast or East Asia”. Also in the material and methods and results section it is not clear in which step this selection is performed. It is not clear from Fig1 what criteria were used to exclude 2641 records.
Results – figure 2 It is not clear what the authors mean with “region of origin”. For example for these ten cases from Canada – the four patients traveled from Southeast Asia? But this is contrary to the statement that only 9 patients had travel history. Perhaps the authors mean that patients emigrate from Asia to the countries of the Americas? Please explain more clearly.
What does prmpA mean – please include explanation, what is difference with rmpA?
It would be good to include analyzes of antimicrobial susceptibility data – it is interesting to show the relationship between the susceptibility of isolates and their virulence
In Figure 3, in the column "microbiological testing, the gene name is difficult to read"
Reviewer 3 Report
Comment Authors may consider additional data analysis and narrative todiscus where is the primary source of infection?
1. Is the liver abscess a primary infection?
2. or is dissemination from a different organ system via bacteremia etc...seeding to liver?
3. Perhaps the GI tract is a main source ( a colonized niche leaking chronically KP )
or alternative is the lung or urinary tract the source of dissemination ?
4. Some of such info is perhaps available for analysis in the authors database and should be considered.
Round 2
Reviewer 2 Report
The authors satisfactorily answered the comments.
Author Response
We are very thankful to the Editor and Reviewers for reviewing our manuscript again.
Please find our responses describing the changes and modifications made to the manuscript below. We believe we have responded satisfactorily to the reviewers edits and suggestions and feel these changes strengthen our manuscript.
Reviewer 3 Report
Authors responded to some of reviewer questions and modified text appropriately.
However, authors did not attempt to formulate working hypothesis which would explain pathogenesis.
I have noted that authors included editorial modifications, virulence factors and mentioned capsular super viscosity features of the pathogens, but this remains just a listing of factors not pathogenesises.
There is still lack of working hypothesis.
Is it possible that properties of KP capsular material interfere with phagocytic properties of liver cells that specialize in killing such pathogen in the environment of diabetes?
Unswearing such question would help general reader and perhaps better target treatment and/or diagnosis.
Authors stated that KP spread occurs via oro-gastric rout but 60% of patients showed bacteremia, so is this a spread from liver abscess or a translocation from GI tract to the blood stream?
This reviewer recognizes that these are difficult questions but directing readers attention towards understanding even hypothetical would be highly relevant.
Author Response
We are very thankful to the Editor and Reviewers for reviewing our manuscript again. Please find our responses describing the changes and modifications made to the manuscript below. We believe we have responded satisfactorily to the concerns and feel these changes strengthen our manuscript.
REVIEWER
Authors responded to some of reviewer questions and modified text appropriately.
However, authors did not attempt to formulate working hypothesis which would explain pathogenesis.
I have noted that authors included editorial modifications, virulence factors and mentioned capsular super viscosity features of the pathogens, but this remains just a listing of factors not pathogenesis.
There is still lack of working hypothesis.
Is it possible that properties of KP capsular material interfere with phagocytic properties of liver cells that specialize in killing such pathogen in the environment of diabetes?
Answering such question would help general reader and perhaps better target treatment and/or diagnosis.
Authors stated that KP spread occurs via oro-gastric rout but 60% of patients showed bacteremia, so is this a spread from liver abscess or a translocation from GI tract to the blood stream?
This reviewer recognizes that these are difficult questions but directing readers attention towards understanding even hypothetical would be highly relevant.
ANSWER
We thank the reviewer for their critical evaluation and comments of our manuscript.
Even though we agree that answering that questions is highly relevant, unfortunately, with our available data (and review of the literature) there is no definitive conclusion that eloquently explains the pathogenesis of Klebsiella liver abscess.
We included an additional paragraph to address these suggestions in the discussion section as follows: ‘The spread occurs most likely through the fecal-oral route, since Klebsiella spp. capsular antigens are known to facilitate intestinal colonization, and clonal dissemination of K1 serotypes has already been described in Korea [31-32]. Nonetheless, the definitive route of entry is still not well-described. It remains uncertain if the pathogen disseminates through the systemic circulation and seeds in the liver, if, conversely, the liver is the primary source of infection (due to translocation from the gastrointestinal tract) from where it disseminates to the blood and other organs, or if it is a combination of both. Regardless, no evidence of gastrointestinal, respiratory or urinary colonization of Klebsiella spp was reported in our cohort.’ (Page 7, paragraph 2)
Also, we listed this gap in knowledge among the limitations of our study: ‘There is still insufficient evidence to elucidate the pathogenesis of cKLA. Additional studies are necessary to generate and test hypotheses in this topic.’ (Page 8, last paragraph.)
Lastly, in light of the recent changes and additions to the discussion, we changed the order of ideas to facilitate the fluidity in the read. No changes in the content were made. Only factual words were changed to allow adequate connection between sentences.